# EVCTRL: EFFICIENT CONTROL ADAPTER FOR VISUAL GENERATION

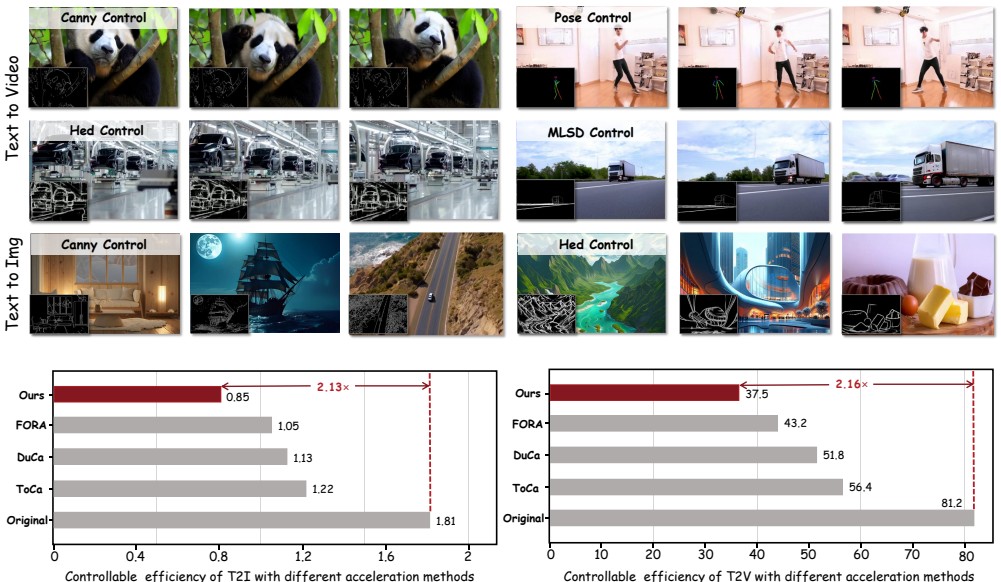

Figure 1: **Showcase of EVCtrl.** We propose the EVCtrl, a lightweight, plug-and-play control adapter. We present the outcomes of controllable image and video generation under various control conditions, along with an efficiency comparison.

## ABSTRACT

Visual generation includes both image and video generation, training probabilistic models to create coherent, diverse, and semantically faithful content from scratch. While early research focused on unconditional sampling, practitioners now demand controllable generation that allows precise specification of layout, pose, motion, or style. While ControlNet grants precise spatial-temporal control, its auxiliary branch markedly increases latency and introduces redundant computation in both uncontrolled regions and denoising steps, especially for video. To address this problem, we introduce **EVCtrl**, a lightweight, plug-and-play control adapter that slashes overhead without retraining the model. Specifically, we propose a spatio-temporal dual caching strategy for sparse control information. **For spatial redundancy**, we first profile how each layer of DiT-ControlNet responds to fine-grained control, then partition the network into global and local functional zones. A locality-aware cache focuses computation on the local zones that truly need the control signal, skipping the bulk of redundant computation in global regions. **For temporal redundancy**, we selectively omit unnecessary denoising steps to improve efficiency. Extensive experiments on CogVideo-Controlnet, Wan2.1-Controlnet, and Flux demonstrate that our method is effective in image and video control generation without the need for training. For example, it achieves $2.16\times$ and $2.05\times$ speedups on CogVideo-Controlnet and Wan2.1-Controlnet, respectively, with almost no degradation in generation quality.Codes are available in the supplementary materials.

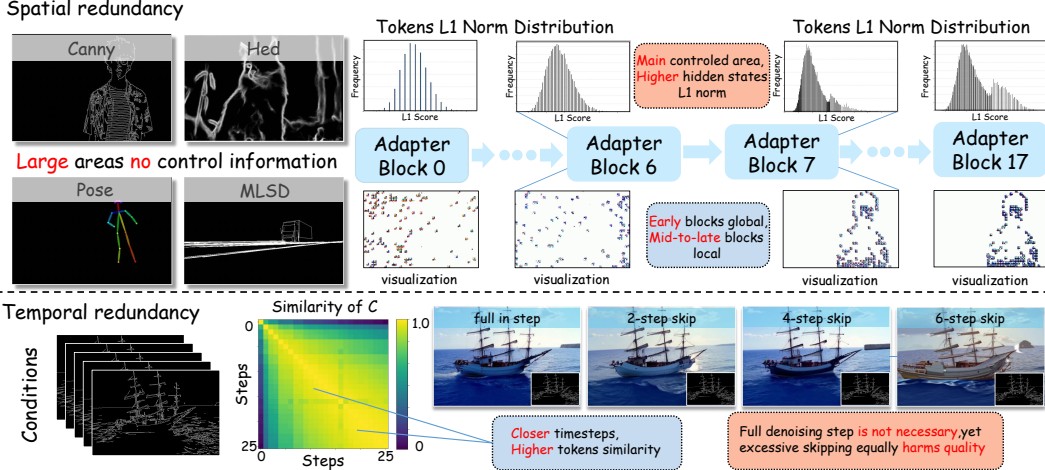

Figure 2: **Motivation illustration.** We observe significant spatial redundancy (a) and temporal redundancy (b) in controllable image and video generation. Spatially(a), large regions in the control image or video that carry no control information need not be computed. Temporally(b), skipping most highly similar adjacent diffusion timesteps does not impair controllability.

# 1 INTRODUCTION

Recent advances in the alignment of machine learning models for text-to-image Ma et al. (2025c; 2023) and text-to-video synthesis Ma et al. (2025a;b;d; 2024d) have propelled a new wave of high-performance generators. Image models such as PixArt-$\delta$Chen et al. (2024a), Stable Diffusion 3Esser et al. (2024), FluxBlack Forest Labs (2024), KolorsKolors Team (2024), and Hunyuan-DiTLi et al. (2024), together with video models including SoraLiu et al. (2024b), CogVideoXYang et al. (2024), HunyuanVideoKong et al. (2024), and Qihoo-T2XWang et al. (2024c), now deliver stunning visual fidelity and open up a wide spectrum of downstream applications.

However, current ControlNet-basedZhang et al. (2023) controlled generation methods for DiT have two major drawbacks. First, as an "auxiliary" neural network model structure, ControlNet typically directly replicates half of the network architecture, such as PixArt-$\delta$Chen et al. (2024a), where a large number of additional parameters significantly increase the inference burden. Similarly, although OminiControlTan et al. (2025) only adds a limited number of parameters, it doubles the number of tokens involved in the attention and linear layers, resulting in an almost 70% increase in overall computational complexity. Second, during the entire diffusion process, the correlation between different time steps and the feature changes of conditional tokens is often overlooked, leading to low computational efficiency. To address these issues caused by computational complexity, this paper first examines the redundancy of conditional information in time and space:

**Spatial Redundancy:** We found that many spatial regions of conditional information are effectively empty, such as edgeless regions whose pixel values sit close to zero. Tokens produced in these zones offer almost no guidance, yet the model still processes them, driving up computation without improving control fidelity. This inefficiency becomes most acute when the condition itself is sparse.

**Temporal Redundancy:** Figure 2 shows the distribution of the feature distances between adjacent time steps for different tokens, where a higher value indicates a lower similarity of the token between adjacent time steps. Observation reveals that conditional tokens C exhibit extremely high distances at most time steps during the diffusion process, while showing relatively low distances at individual time steps. This observation suggests that there is more significant temporal redundancy in Controlnet.

Motivated by these observations, we propose an **E**fficient **V**isual **C**ontrol Adapter(**EVCtrl**). For spatial redundancy, EVCtrl presents **L**ocal **Fo**cused **C**aching(**LFoC**). LFoC identifies and stores only those tokens whose caching yields the highest benefit-to-cost ratio, prioritizing regions that encode salient edge cues. Concretely, during the denoising process, the adapter first computes and caches tokens across all spatial locations. In subsequent steps, tokens corresponding to edges are recom-

puted in full to preserve fidelity, whereas tokens associated with near-zero, edge-free regions are retrieved from the cache. This hybrid strategy enables the diffusion model to concentrate its capacity on edge-conditioned areas, sustaining control quality while substantially reducing computation over uniform regions.

Additionally, we observe temporal redundancy within the denoising processing: the conditioning signal evolves unevenly across timesteps. EVCtrl's **D**enoising **S**tep **S**kipping (**DSS**) quantifies the dynamic variation of the condition at each step and retains full computation for the few timesteps that contribute most to the final control. By concentrating resources on these pivotal steps, DSS simultaneously lowers computational load and improves output quality. Numerous experiments in text-to-image and text-to-video generation have shown that EVCtrl is more effective than previous feature caching methods on CogVideo-ControlnetTheDenk (2024), Wan2.1-Controlnet, and Flux-ControlnetXLabs-AI (2024). For example, on CogVideo-ControlnetTheDenk (2024), it can achieve$2.16\times$ acceleration, preserving near-lossless quality across various evaluation metrics when compared to the original controlnet pipeline without cache acceleration.In general, our contributions can be summarized as follows:

- We emphasize the spatial and temporal redundancies inherent to image and video generation tasks, and propose EVCtrl, a novel and efficient control adapter for controllable image and video generation.

- **Local Focused Caching (LFoC)**: To reduce spatial redundancy in the control signal, we propose spatial locality caching, which skips most computations over regions that receive no control . Then we identify the key carriers of fine-grained control information within DiT-ControlNet and construct a hierarchical attribution–sensitivity framework to further accelerate caching.

- **Denoising Step Skipping (DSS)**: To reduce temporal redundancy, we exploit the higher redundancy of the control branch relative to the original denoising path and the uneven influence of individual denoising steps on the conditioning signal, adaptively selecting the few steps that most affect the control for full computation while maintaining a periodic cache.

- Extensive experiments have been conducted on CogVideo-Controlnet, Wan2.1-Controlnet, and Flux-Controlnet, and the results show that EVCtrl achieves a high acceleration ratio while maintaining nearly lossless generation quality.

## 2 RELATED WORKS

### 2.1 CONTROLLABLE DIFFUSION MODELS

To achieve conditional control in pre-trained text-to-image diffusion models, there are two main methods for introducing controllable conditions into image Yan et al. (2025); Xiong et al. (2025); Yuluo et al. (2025b); Zhang et al. (2025b); Zhu et al. (2024) or video generation models Ma et al. (2024c; 2022): (1) training a new large model from scratch for multi-condition compliance Huang et al. (2023) (2) freezing the large pretrained model and fine-tuning only a lightweight adapter. Recent studies have extended these ideas to diffusion transformers (DiTs) Tan et al. (2024); Wang et al. (2024a); Cai et al. (2025); Chen et al. (2024b); Cao et al. (2025); Wan et al. (2024); Feng et al. (2025b); Mao et al. (2025); Yuluo et al. (2025a); Li et al. (2025); She et al. (2025), exploiting the built-in multimodal attention (MM-attention)Pan et al. (2020) to inject image or video conditions without further architectural change. T2I-AdapterMou et al. (2024) exemplifies the adapter route—it imposes minimal overhead yet yields weak control, limiting its use on complex prompts. In contrast, ControlNetZhang et al. (2023) achieves stronger control effects by copying specific layers from pre-trained large models to guide image generation to align with control information, but it introduces a large number of additional parameters. Meanwhile, as the number of condition tokens increases, especially in controllable video generation, performing self-attention operations on the entire sequence of condition tokens leads to a significant increase in computational cost, imposing significant limitations on the controllable generation result rate from text to video.

## 2.2 ACCELERATION OF DIFFUSION MODELS

Implementing low-latency and high-quality generation methods on diffusion models (DMs), including DiT, is an important research direction. Currently, there are mainly two types of acceleration methods for diffusion models: the first type is to reduce the number of sampling stepsLiu et al. (2022); Wang et al. (2024b); Lu et al. (2022); Zhu et al. (2025a); Chen et al. (2023); Lu et al. (2025); Song et al. (2020), and the second type is to accelerate the internal computation of diffusion models. Solutions to reduce computational complexity include model distillation and compressionXie et al. (2024), token mergingBolya et al. (2022); Bolya & Hoffman (2023); Feng et al. (2025a), token pruningZhang et al. (2025a), and layer-wise caching techniquesGao et al. (2024); Xiao et al. (2023); Chen et al. (2024c); Liu et al. (2024a); Ma et al. (2024b;a); Selvaraju et al. (2024). However, layer-wise caching techniques use entire layers as the granularity and have difficulty perceiving the importance differences between tokens, resulting in key information being equally frozen or discarded. TocaZou et al. (2024a) and DucaZou et al. (2024b) instead use tokens as the minimum caching unit, accurately measure, and explicitly assign importance weights to each token with the help of a score map. In the subsequent computation stage, only high-weight tokens are refreshed according to a predetermined ratio, and the rest are directly reused from the cache, thus achieving lossless acceleration.

Unfortunately, the existing caching schemes are not specifically designed to adapt to ControlNet, which replicates specific layers in DM, and have the following issues: First, the existing caching and editing schemes often rely on explicit read and write operations of attention mapsZou et al. (2024a) or KV matricesLu et al. (2023); Zhu et al. (2025b), which are incompatible with the mainstream Transformer acceleration pipeline and instead introduce additional latency. Additionally, there are differences in the correlation and importance of control information among different transformer layers in the existing diffusion transformer controlnetCao et al. (2025). Meanwhile, the control conditions themselves provide prior information on the spatial distribution that should be focused on, but current methods fail to utilize these differences and information.

## 3 METHOD

### 3.1 OVERALL FRAMEWORK

The pipeline of our method is show in Figure 3.The core idea of EVCtrl is that, within each regular caching cycle, it performs full computation on an additional handful of denoising steps whose latent states critically determine the final output, while continuously updating the tokens that encode fine-grained control information, thereby mitigating errors introduced by repeatedly reusing stale control-signal features.In addition, existing caching strategies in deployed pipelines only cache the intermediate features of the MLP sub-layers inside a Transformer block. Unfortunately, the attention outputs themselves accumulate progressively larger errors when their outdated versions are reused, and these errors propagate downstream. Therefore, we extend the cache-update procedure to refresh the feature maps in both the attention and the MLP sub-layers simultaneously.

In the following, we first introduce the Spatial Locality Focused Caching to tackle the spatial redundancy within the Controlnet branch. Then we introduce the Temporal Denoising Step Skipping to tackle the temporal redundancy within the same branch.

### 3.2 SPATIAL LOCALITY FOCUSED CACHING

#### 3.2.1 LAYER SENSITIVITY IN DIT-CONTROLNET

To systematically characterise the layer-wise influence of DiT-ControlNet on both perceptual fidelity and conditional control accuracy, we propose a hierarchical attribution and sensitivity quantification framework that proceeds in three conceptually linked stages: norm-based observation, semantic mapping, and functional stratification. In the first stage, we extract every spatial token from the feature tensor produced by each DiT-ControlNet layer and compute its $L1$ norm. Global distributional properties are then estimated via histograms and kernel-density curves. In the second stage, a structure-similarity-driven alignment mechanism back-projects the indices of high-norm tokens into the original conditional domain, measuring their spatial overlap with the provided edge or

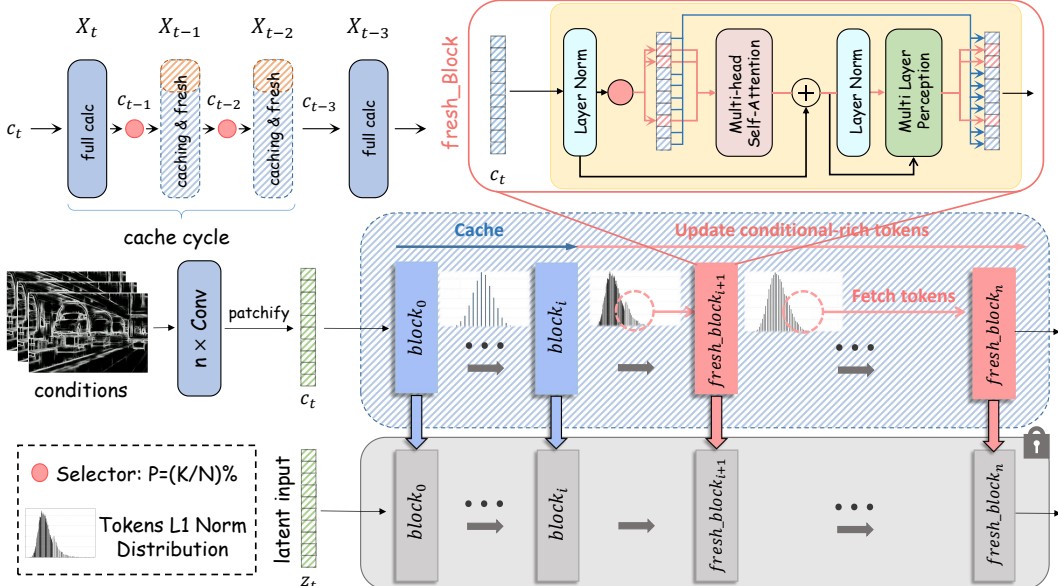

Figure 3: **Overall framework**.The pipeline takes initial noise and control conditions as inputs. Specifically, full caches are refreshed at fixed timestep intervals and at a handful of pivotal timesteps. For cached intermediate steps, only selected tokens within the mid-to-late fresh_blocks that capture local salient control cues are updated.

pose guidance. This establishes an explicit correspondence between activation magnitude and fine-grained control semantics. Guided jointly by these two indicators, the network is stratified into two functional zones: (1) early layers exhibit peaked, low-variance distributions and are interpreted as robust encoders of low-frequency, global structural priors (2) mid-to-late layers display heavy-tailed distributions in which a minority of tokens obtain disproportionately large activations, thereby capturing high-frequency, local control cues such as edges, corners, and texture discontinuities. These high-energy tokens are identified as the primary carriers of fine-grained control information. This stratification paradigm is entirely training-free, providing an interpretable theoretical basis for subsequent selective caching strategies.

### 3.2.2 SPATIAL TOKEN-WISE FEATURE CACHING

Analogous to prior token-wise conservative caching schemes, ToCa offers two attention map-based selection strategies, whereas Duca directly retains tokens whose value matrices exhibit small norms as surrogates for those receiving high attention scores, thereby neutralizing the incompatibility between attention-score-driven token selection and mainstream attention optimizers such as FlashAttention and other memory-efficient variants. However, these two token selection paradigms remain insufficiently tailored to the DiT-ControlNet architecture; therefore, we propose a markedly superior alternative to select conditional tokens.

Figure 2 empirically shows that tokens possessing large norms are strongly correlated with edge or pose conditioning signals.This observation motivates a caching strategy. It refreshes high-norm tokens more often within each interval. These tokens belong to mid-to-late layers that capture high-frequency local control cues and carry fine-grained information, thus strengthening conditional guidance. The entire procedure can be inserted into any DiT-ControlNet inference pipeline without parameter modification or retraining, realizing fine-grained caching that "refreshes frequently where critical and sparingly where redundant." Concretely, let $P$ denote the proportion of critical conditional information; hence, the selection rule for $\mathcal{P}_{Cache}$ is defined as follows:

$$\mathcal{P}_{Cache} = \underset{\{i_1, i_2, \ldots, i_n\} \subseteq \{1, 2, \ldots, N\}}{\arg\max} \{\|t_{i_1}\|_1, \|t_{i_2}\|_1, \ldots, \|t_{i_n}\|_1\} \tag{1}$$

where the $n$ is the total number of selected tokens, defined by $n = \lfloor P\% \times N \rfloor$

Table 1: **Quality and efficiency benchmarking results** of EVCtrl and other baselines

| Type | Method | Quality | | | | | Efficiency | |
|------|--------|---------|------|------|-------|--------|---------|---------|
| | | FID↓ | PSNR↑ | SSIM↑ | LPIPS↓ | VBench↑ | Latency↓ | Speedup↑ |
| **T2I** | Flux-ControlNet | – | – | – | – | – | 1.81s | 1× |
| | + Fora(N=2) | 47.97 | 27.65 | 0.92 | 0.053 | – | 1.05s | 1.72× |
| | + Toca(N=2) | 55.82 | 26.98 | 0.90 | 0.065 | – | 1.22s | 1.48× |
| | + Toca(N=4) | 64.78 | 25.43 | 0.88 | 0.134 | – | 1.09s | 1.66× |
| | + Taylorseer(N=2,O=4) | 92.13 | 24.56 | 0.84 | 0.182 | – | 0.97s | 1.87× |
| | **+ Ours(N=4)** | **34.69** | **28.74** | **0.94** | **0.042** | – | **0.93s** | **1.95×** |
| | + Fora(N=4) | 56.73 | 26.12 | 0.89 | 0.101 | – | 0.87s | 2.08× |
| | + Taylorsser(N=4,O=4) | 148.71 | 18.44 | 0.76 | 0.249 | – | 0.82s | 2.21× |
| | **+ Ours(N=8)** | **42.34** | **27.88** | **0.91** | **0.057** | – | **0.85s** | **2.13×** |
| **T2V** | CogVideoX-ControlNet | – | – | – | – | 82.6% | 81.2s | 1× |
| | + Fora(N=2) | 51.45 | 23.55 | 0.83 | 0.13 | 82.2% | 49.5s | 1.64× |
| | + Toca(N=2) | 62.97 | 20.93 | 0.78 | 0.18 | 81.6% | 57.8s | 1.40× |
| | + Toca(N=4) | 71.34 | 18.84 | 0.75 | 0.21 | 79.8% | 49.2s | 1.61× |
| | + Taylorseer(N=2,O=4) | 108.36 | 17.89 | 0.67 | 0.26 | 76.7% | 46.4s | 1.75× |
| | **+ Ours(N=4)** | **51.37** | **23.62** | **0.87** | **0.11** | **82.2%** | **43.4s** | **1.88×** |
| | + Fora(N=4) | 66.98 | 21.73 | 0.74 | 0.17 | 80.6% | 40.1s | 2.02× |
| | + Taylorsser(N=4,O=4) | 160.48 | 14.97 | 0.53 | 0.38 | 72.9% | 35.0s | 2.32× |
| | **+ Ours(N=8)** | **64.83** | **22.13** | **0.81** | **0.15** | **81.3%** | **38.5s** | **2.16×** |

## 3.3 TEMPORAL DENOISING STEP SKIPPING

### 3.3.1 DENOISING STEP REDUNDANCY AND CRITICALITY DETECTION

After addressing spatial redundancy, we focus on the temporal redundancy that naturally emerges during iterative denoising in the control branch. As shown in Figure 2, we pairwise compare latent vectors across all timesteps through cosine similarity. Strikingly, tokens produced in consecutive denoising steps exhibit near-identical similarities, indicating an extremely slow evolution in latent space. This near-uniform behavior implies that substantial computational resources are repeatedly expended on nearly identical representations. Meanwhile, although most adjacent denoising steps are highly redundant, Figure 2 also reveals abrupt drops in similarity at isolated timesteps. These sudden divergences mark rapid pivots in the denoising trajectory and inject details critical to final perceptual quality; consequently, these critical steps exert a disproportionately large influence on the final generation.

### 3.3.2 STRIDE-SKIPPING WITH CRITICAL-STEP PRESERVATION

To take advantage of the inherent similarity between adjacent time steps more efficiently, without altering the underlying model architecture, we set a cache interval $N$. Features are fully recomputed and cached every $N$ steps, then reused for the subsequent $N-1$ timesteps. To prevent overly aggressive skipping from inadvertently discarding steps that significantly affect quality, we introduce a complementary mechanism: within each cache interval, critical steps identified a priori are selectively restored to full computation, ensuring essential details are preserved.Concretely, given a set of $N$ adjacent timesteps$\{t, t-1, \ldots, t-(N-1)\}$,we execute a full forward pass at the cycle's leading timestep $t$ to compute and store the features in a cache as $C_t^l := F(x_t^l)$,where the superscript $l \in \{1, 2, \ldots, L\}$ denotes the layer index. Additionally, for a predetermined sequence of critical steps$\{k_0, k_1, \ldots k_m\}$,we assume a timestep $k_i$ satisfies $t-1 \leq k_i \leq t-(N-1), 0 \leq i \leq m$.A full forward computation is executed at $k_i$ to update the cache $C_{k_i}^l := F(x_{k_i}^l)$. The cached representation is then reused for all remaining redundant timesteps so that $F(\cdot)$ is not invoked again.

Table 2: **Quality and efficiency benchmarking results** of EVCtrl and other baselines across diverse control conditions

| Method | Canny | | | | | Hed | | | | |
|---|---|---|---|---|---|---|---|---|---|---|
| | Speed↑ | FID↓ | PSNR↑ | SSIM↑ | LPIPS↓ | Speed↑ | FID↓ | PSNR↑ | SSIM↑ | LPIPS↓ |
| + Fora(N=4) | 1.98× | 64.85 | 22.74 | 0.75 | 0.16 | 1.97× | 64.67 | 22.56 | 0.75 | 0.17 |
| + Toca(N=4) | 1.55× | 69.83 | 21.32 | 0.75 | 0.19 | 1.55× | 69.74 | 21.25 | 0.76 | 0.18 |
| + Taylorseer(N=2) | 1.85× | 104.68 | 19.94 | 0.69 | 0.24 | 1.86× | 104.59 | 19.92 | 0.69 | 0.25 |
| + **Ours(N=8)** | **2.05×** | **62.24** | **23.26** | **0.80** | **0.12** | **2.02×** | **63.74** | **22.84** | **0.78** | **0.14** |

| Method | Pose | | | | | MLSD | | | | |
|---|---|---|---|---|---|---|---|---|---|---|
| | Speed↑ | FID↓ | PSNR↑ | SSIM↑ | LPIPS↓ | Speed↑ | FID↓ | PSNR↑ | SSIM↑ | LPIPS↓ |
| + Fora(N=4) | 1.98× | 64.82 | 22.63 | 0.74 | 0.17 | 1.98× | 64.79 | 22.68 | 0.75 | 0.16 |
| + Toca(N=4) | 1.56× | 69.81 | 21.31 | 0.75 | 0.19 | 1.55× | 69.75 | 21.46 | 0.75 | 0.18 |
| + Taylorseer(N=2) | 1.86× | 104.47 | 19.86 | 0.70 | 0.22 | 1.87× | 104.62 | 19.89 | 0.69 | 0.24 |
| + **Ours(N=8)** | **2.23×** | **56.64** | **24.76** | **0.82** | **0.09** | **2.08×** | **60.31** | **23.83** | **0.81** | **0.11** |

## 4 EXPERIMENTS

### 4.1 EXPERIMENTAL SETUPS

#### 4.1.1 MODEL CONFIGURATIONS

We conducted experiments with three commonly used DIT-based models equipped with Control-Net for various generative tasks, including FluxXLabs-AI (2024) for text-to-image generation, CogVideoTheDenk (2024), and Wan2.1Wan et al. (2025) for text-to-video generation. The experiments were performed using an NVIDIA A800 80GB GPU. The CogVideo5B model is capable of generating 6-second videos with a resolution of 480p and a frame rate of 8fps. The Wan2.1-14B model can generate videos with a resolution of 720p or 480p, a frame rate of 16fps, and a duration of 5 seconds. Each model used its default sampling method: DPM-Solver++Lu et al. (2025) with 20 steps for Flux, and DDIMSong et al. (2020) with 50 steps for both CogVideo and Wan2.1. For each model, we configure EVCtrl by setting the cache ratio $P$ according to the redundancy level of the conditional information, while also employing various average forced-activation periods $N$.

#### 4.1.2 EVALUATION PROTOCOL AND METRICS.

We randomly selected 5,000 images and captions from the COCO-2017Lin et al. (2014) dataset and extracted 100 videos and their descriptions from each of the 16 distinct categories of videos in the VbenchHuang et al. (2024) dataset, totaling 1,600 videos. We evaluated the quality of the generated content using SSIMWang et al. (2004), LPIPSZhang et al. (2018), PSNRWang et al. (2004), and FIDHeusel et al. (2017). Additionally, for text-to-video generation, we used video generation metrics within the Vbench framework as supplementary indicators.

### 4.2 COMPARISON WITH BASELINES

#### 4.2.1 QUANTITATIVE RESULTS

As shown in Table 1, we benchmark EVCtrl against three state of the art training free acceleration baselines: ForaSelvaraju et al. (2024), TocaZou et al. (2024a), and TaylorseerLiu et al. (2025), on both the Flux-ControlNetXLabs-AI (2024) and CogVideo-ControlNetTheDenk (2024) pipelines. Quantitative results demonstrate that EVCtrl outperforms all other baselines across every quality metric (FID ↓, PSNR ↑, SSIM ↑, LPIPS ↓) while achieving equal or even higher acceleration ratios. Specifically, on the COCO2017 validation set at a $2.13\times$ speedup, EVCtrl achieves the lowest FID of 42.34 and maintains SSIM above 0.9; on the VBench validation set, it delivers a $2.16\times$ speedup with SSIM = 0.81, and all remaining metrics also surpass those of competing baselines at the same speedup, without any retraining or extra memory cost.

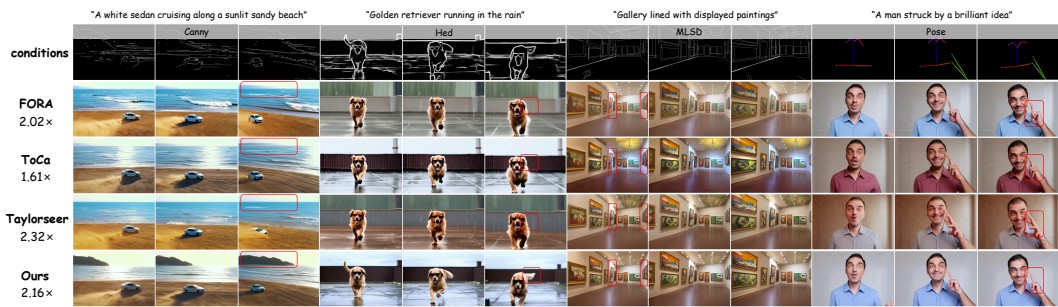

Figure 4: **The qualitative comparisons with existing methods.** Visualization comparing acceleration methods on Wan2.1-ControlNet: while others sacrifice control-condition details and introduce visual distortion at high speed-up ratios, ours preserves fine-grained details and maintains high-quality generation.

It is also noteworthy that as the caching period lengthens and the overall acceleration ratio continues to climb, the baseline methods suffer a sharp quality drop due to the loss of critical control signals during strided propagation: FID and LPIPS deteriorate rapidly, while PSNR and SSIM decline in tandem, leading to obvious artifacts or structural distortions in the generated images. By contrast, EVCtrl maintains exceptional adaptability; even when the caching period increases from $N = 4$ to $N = 8$, it still exhibits an unrivaled balance between efficiency and fidelity.

In addition, as summarized in Table 2, we evaluated EVCtrl in the Wan2.1-ControlNet pipeline under four distinct control modalities: Canny, HED, OpenPose, and MLSD. Quantitative results reveal that, regardless of the type of control, EVCtrl consistently outperforms all baselines on every quality metric (FID ↓, PSNR ↑, SSIM ↑, LPIPS ↓) while achieving equal or greater end-to-end acceleration. In particular, as the control signal becomes sparser, e.g., the thin edge maps produced by Canny or the minimal keypoint representation of OpenPose, the acceleration ratio of EVCtrl rises further, reaching up to $2.4\times$ on OpenPose without any quality degradation. This trend confirms that EVCtrl is especially effective when the conditioning information is low-density, enabling it to skip redundant computation while still preserving fine-grained structural fidelity.

Table 3: **Ablation study on different configurations of LFoC and DSS** for controllable video generation

| Method | | Latency ↓ | SSIM ↑ | LPIPS ↓ |
|---|---|---|---|---|
| **LFoC-only** | WA | 37.6s | 0.78 | 0.19 |
| | WM | 36.6s | 0.74 | 0.22 |
| **DSS-only** | | 35.8s | 0.66 | 0.27 |
| **EVCtrl** | | 39.2s | 0.81 | 0.15 |

### 4.2.2 QUALITATIVE RESULTS

We compare EVCtrl with prior accelerators on Wan2.1-ControlNet under four control conditions (Canny, HED, MLSD, pose), highlighting its ability to accelerate inference while preserving controllable detail and video quality. The results are shown in 4. Across all conditions, EVCtrl consistently outperforms others in fidelity and detail: under Canny, it accurately reconstructs the distant mountains in the scene "a white sedan cruising along a sunlit sandy beach"; under HED, it retains the golden retriever's tail in "golden retriever running in the rain"; under MLSD, it best matches the structural conditions in "gallery lined with displayed paintings"; and under pose guidance, the human hands exhibit both high quality and fidelity. In summary, EVCtrl leads existing methods in fidelity and detail.

| LFoC-only-WM | LFoC-only-WA | DSS-only | EVCtrl |

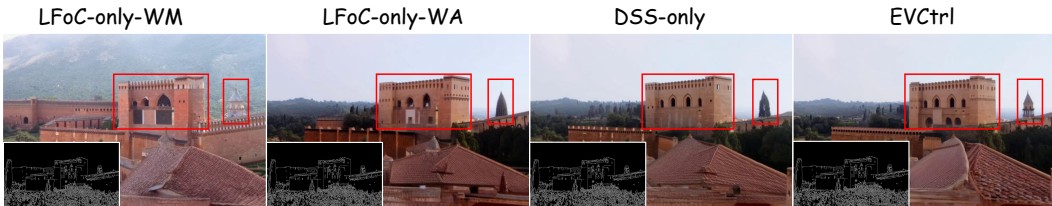

Figure 5: **The qualitative ablation study.** We select the canny as the input condition to ablate the performance.

### 4.3 ABLATION STUDY

In the subsequent section, we will analyze the effectiveness of spatial module LFoC and temporal module DSS.

#### 4.3.1 EFFECTIVENESS OF LOCAL FOCUSED CACHING(LFoC)

Table 3 shows that removing LFoC lowers SSIM from 0.81 to 0.66 and raises LPIPS from 0.15 to 0.27, indicating a decline in generation quality. Moreover, comparing LFoC-with-attention(WA) and LFoC-with-MLP(WM) reveals that updating either the attention layers alone or the MLP layers alone leaves the cached key control tokens under-utilized. Only when LFoC refreshes both the attention and MLP layers simultaneously can the re-injected tokens fully unleash their expressive power, allowing the model to recover its original quality metrics. The visual results in Figure 5 demonstrate that after removing LFoC the model struggles to control fine details, illustrating that LFoC better exploits spatially local, fine-grained prior knowledge and achieves superior controllability in controlled generation tasks compared with the aggressive cache approach and the approach Zou et al. (2024a)Zou et al. (2024b) that only updates the MLP layers.

#### 4.3.2 EFFECTIVENESS OF DENOISING STEP SKIPPING(DSS)

As shown in Table 3, DSS effectively shortens inference time with almost no loss in quality.Additionally, Table 1 demonstrates that DSS allows the skip interval $N$ to be increased to 8 for reduced computational cost while maintaining the same or better controllable-generation performance as other acceleration methods achieve at only half that skip interval. Moreover, the visual results in Figure 5 reveal that removing DSS leads to a marked decline in both overall scene completeness and the fidelity of detail control, further validating the effectiveness and fidelity of our DSS strategy.

## 5 CONCLUSION

ControlNet-based controllable image and video generation usually incurs high computational costs due to spatial and temporal redundancies in diffusion models.To address this problem, we present EVCtrl, a training-free plug-and-play adapter that focuses computation on spatially local control-relevant tokens through spatial Local Focused Caching and prunes redundant denoising iterations through temporal Denoising Step Skipping. Extensive results demonstrate EVCtrl's effectiveness for efficient controllable generation under diverse conditions. Our work offers valuable insights and directions toward future efficient, potentially real-time controllable image and video generation.

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

# A APPENDIX

## A.1 PROMPT DETAILS

In Figure 1, the prompts for the videos and images are arranged top-to-bottom, left-to-right as follows: (1)"A cute panda sits on a tree branch" (2)"A young man is joyfully dancing in his room"(3)"Bright high-tech factory with automated electric-car production line"(4)"A freight truck cruises along an international highway"(5)"Scandinavian cozy interior, pale ash wood and warm candle-lit cream tones, ultra-HD, cinematic mood"(6)"Moonlit galleon sailing beneath bright moonlight"(7)"A white sedan cruises along a cliff-hugging coastal road"(8)"Masterpiece, best_quality, absurdres, conceptual art, landscape, mountains, fantasy, waterfalls, green, lush, valley, dramatic, artistic, illustration, nature, rocks, clouds, mystical, vibrant, aerial view, beach, forest, ocean, coastline, tropical, sand, waves, blue water, green trees, drone shot"(9)"Futuristic 500m shopping tower, glass curves, warm lights, night city"(10)"On the table sit chocolate, butter, bread, eggs, and a jug of milk"

In Figure 2, the video prompt is: "A majestic three-masted sailing ship is sailing across the vast ocean under a bright and sunny sky."

In Figure 5, the video prompt is:"The Alhambra in daylight."

## A.2 POWERPOINT FOR VIDEO PRESENTATION

To further demonstrate the superiority of EVCtrl in video generation, we have prepared an anonymous PowerPoint presentation. Please refer to the PowerPoint file in the Supplementary Materials for more detailed viewing.

