# OpenReview forum: "EVCtrl: Efficient Control Adapter for Visual Generation"
_ICLR.cc/2026/Conference — ICLR 2026 Conference Withdrawn Submission_

### Official Review · Reviewer_CycW · 2025-10-25

**Soundness:** 3
**Presentation:** 2
**Contribution:** 2
**Rating:** 4
**Confidence:** 4

**Summary:**

The authors propose EVCtrl, a training free method for speeding up inference of ControlNet based models. The method is based on the observation that when using sparse conditioning modalities mostly consisting of black pixels, only tokens corresponding to non-black regions are critical for updating. In addition, authors identify precise steps along the denoising trajectory where large changes happen inside the model, requiring full recomputation of features to preserve performance. Evaluation on CogVideo, Flux and Wan show positive performance and better or comparable speedups with respect to the baselines on the chosen metrics.

**Strengths:**

- The method achieves higher speedup and better metrics with respect to the chosen baselines
- The small amount of showcased qualitative results shows some cases where the method is successful while baselines are failing.
- The analysis of token magnitudes, correlating high magnitude tokens to contour regions is interesting

**Weaknesses:**

- The amount of provided qualitative results in the form of videos is limited to 6 samples in the supplementary power point presentation, making it difficult to qualitatively judge the quality of the method.
- The method relies on "Local Focused Caching (LFoC) to achieve its speedup and performance. LFoC relies on the assumption that control signals are sparse and mostly consisting of spatially uninformative regions (e.g. poses, edges). Usage of dense control signals such as depth maps should be considered and discussed, as it is of practical interest but might cause failure of the method. It is currently unclear how the method would perform on these control signals with respect to baselines.
- The method relies on the identification of critical steps, and on the identification of global and local model regions. The procedure needs be repeated for each method, requiring some effort from the user for correct application. The paper provides limited guidance on how this can be achieved in 3.3.2, which might make the presented method difficult to apply in practice.
- LL150-151 only present the case of retraining a large model from scratch, or use of control nets. Recent methods such as VACE achieve strong controllability through model fine-tuning. Several recent methods showcase strong conditioning capabilities through model fine-tuning, reducing significance of the work which is tailored to the specific case of ControlNets.
- Quality of writing is problematic with several typos that disturb reading, e.g. method names fused with author names, missing spaces, incorrect punctuation LL79, 81, 84, 105, 118, 119, 121, 170, 171, 172, 318, 322, 371, 372 ...

**Questions:**

- It would be helpful to prove method significance to extend the evaluation to commonly used dense conditioning modalities such as depth, and show that the proposed method can compare favorably to baselines.
- Could the authors include the original control net model in tab 2 to serve as an upper bound?
- Please specify the unit of measure on the X axis for plots in Figure 1?

**Details Of Ethics Concerns:**

The work contains several citations that either do not constitute the most significant piece of related work, or appear misplaced. An analysis of these citations reveals a recurrent author, named "Yue Ma", whose 25 different works are included as references. 25 out of 69 cited works come from this author, amounting for 36% of the chosen references.

It is difficult to think these 25 works could have been cited in good faith, believing they constitute the most relevant citation for the citing context. I believe "Yue Ma" is one of the paper authors and that these works constitute self citations aimed at boosting its own citation statistics.

If that was the case, such behavior would be indicative of academic misconduct. I suggest an ethics review to be performed on the submission to determine if academic misconduct is present.

---

### Official Review · Reviewer_kwyz · 2025-10-30

**Soundness:** 3
**Presentation:** 3
**Contribution:** 3
**Rating:** 8
**Confidence:** 4

**Summary:**

This paper introduces EVCtrl, an innovative, training-free, plug-and-play acceleration framework that tackles the low efficiency and high computational cost of ControlNet-based controllable image/video generation. By exploiting spatial redundancy via Local-Focused Caching (LFoC) and temporal redundancy via Denoising-Step Skipping (DSS), EVCtrl achieves 2.16 times speed-up on CogVideo-ControlNet without sacrificing quality. Extensive experiments on Flux, CogVideo, and Wan2.1 under four control conditions show superior SSIM, PSNR, LPIPS, and lower latency compared with existing training-free acceleration baselines. However, performance under high-resolution, long-video, or complex control scenarios has not been examined, and comparisons with non-ControlNet controllers are missing; consequently, the generality of the approach remains insufficiently validated. Although the novelty is somewhat conservative and lacks theoretical depth, the paper is of good overall quality. Strengthening theoretical analysis and presentation in the final version is expected to further increase its impact.

**Strengths:**

1. High practicality: zero training cost, plug-and-play deployment.
2. Clear redundancy modeling: separates spatial (LFoC) and temporal (DSS) redundancy.
3. Comprehensive experiments: multiple models and control conditions, significant acceleration.
4. Quality preservation: maintains visual metrics comparable to the original ControlNet under 2.16 times speed-up.

**Weaknesses:**

1. Limited generalization: no tests on high-resolution, long-video, or complex control scenarios.
2. Narrow comparison: only evaluated against training-free acceleration baselines; Insufficient comparison with other types of controllers or acceleration methods.
3. Shallow theoretical analysis: lacks formal discussion on why and when LFoC and DSS work or fail.

**Questions:**

1. Does the local caching window of LFoC and the skipping schedule of DSS remain effective for 2K/4K images or videos longer than 100 frames?
2. When control signals are time-varying or multi-modal, how does DSS ensure consistent denoising steps without introducing flickering or artifacts?
3. Compared with distillation, quantization, or NAS-based acceleration, what is the performance-to-cost ratio and theoretical upper bound of EVCtrl?
4. Can you provide a theoretical or statistical relationship between LFoC cache-hit rate and generation quality degradation?

---

### Official Review · Reviewer_2v8v · 2025-11-01

**Soundness:** 3
**Presentation:** 2
**Contribution:** 3
**Rating:** 4
**Confidence:** 4

**Summary:**

This work proposes EVCtrl, an efficient control adapter mainly addressing the spatial redundancy and temporal redundancy issues prevalent in current image and video generation methods that leverage ControlNet. This method has been experimented on the ControlNet branches of multiple models, achieving impressive acceleration effects without significant visual degradation in generation quality.

**Strengths:**

1. The motivation of this paper is clear, i.e., addressing the temporal and spatial redundancy caused by utilizing ControlNet.
2. The method is training-free and easy to implement, requiring no retraining. The LFoC component attempts to explore the internal working principles of DiT-ControlNet by analyzing the functions of different layers via L1 norm, which is a promising approach to optimize ControlNet.
3. Extensive experiments have been conducted, both qualitatively and quantitatively, consistently demonstrating the effectiveness of EVCtrl.

**Weaknesses:**

1. The DSS (temporal acceleration) mechanism is insufficient explained, lacking crucial details for reproducibility. Terms like "identified a priori" and "predetermined sequence of critical steps" are used without explaining the specific criteria for screening these steps or specifying the quantity $m$. This missing information is vital.
2. The methodology is not very novel. The core idea of DSS is similar to prior work like FORA, where FORA has already verified the effectiveness of exploiting temporal redundancy in DiT through feature caching and reuse. The innovation of DSS is an application-specific migration of this known temporal acceleration strategy to the ControlNet branch, fine-tuned with the empirical design of "critical steps." Similarly, the essence of LFoC lies in token pruning or sparse computation, an approach already established by precursors such as ToCa and Duca.
3.The expression of this paper requires improvement. Furthermore, there are several typos, such as the inconsistent spelling of "Taylorseer" and "Taylorsser" in Table 1. More errors can be found in the Questions section.

**Questions:**

1. Please clarify the discrepancy between Figure 1 T2V latency (37.5s) and Table 1 T2V latency (38.5s). Additionally, in Table 1, all quality metrics (FID, PSNR, etc.) of the "Original" Flux-ControlNet are missing, leaving the primary claim of "no quality degradation" unsubstantiated.
2. Please clarify the relationship between the parameters $P$ and $N$ in Equation (1) and the $P$ and $N$ used in the experiments (L268-270). Re-stating their precise definition is necessary as their role is fundamental to the method.
3. Given the ambiguity of "predetermined critical steps" (Weakness 1), could the authors provide a more concrete, adaptive online selection method for key steps (e.g., based on adjacent step cosine distance or gradient norm)?

---

### Official Review · Reviewer_djAJ · 2025-11-02

**Soundness:** 3
**Presentation:** 3
**Contribution:** 3
**Rating:** 6
**Confidence:** 4

**Summary:**

**Summary:**
This paper proposes **EVCtrl**, a lightweight, training-free control adapter designed to improve the efficiency of controllable image and video generation based on diffusion transformers (DiTs). The method addresses the significant spatial and temporal redundancies in ControlNet by introducing two components: **Local Focused Caching (LFoC)** for spatial redundancy reduction and **Denoising Step Skipping (DSS)** for temporal redundancy pruning. EVCtrl selectively recomputes tokens containing fine-grained control cues while caching redundant ones, and dynamically skips diffusion steps with high similarity between adjacent states. Experiments on multiple image and video diffusion backbones (Flux, CogVideo, Wan2.1) show 2.0×‒2.4× acceleration with minimal degradation in FID, SSIM, and LPIPS scores.

**Strengths:**

**Strengths:**
- Proposes a **clear and practical approach** that effectively targets spatial and temporal redundancy in controllable diffusion models.
- **Training-free and plug-and-play**, making it highly practical for real-world deployment and easily integrated into existing ControlNet and DiT pipelines.
- **Comprehensive experiments** across text-to-image and text-to-video tasks with multiple baselines (Fora, ToCa, Taylorseer) strongly support the claimed acceleration and quality preservation.
- Introduces **interpretable token-sensitivity analysis** and hierarchical feature profiling, helping justify the LFoC design.
- Ablation studies and qualitative visualizations effectively demonstrate the contribution of each module (LFoC, DSS).

**Weaknesses:**

Weaknesses:
1. The paper’s novelty is **incremental**, mainly improving upon existing caching or skipping techniques (e.g., ToCa, Duca) rather than proposing a fundamentally new principle.
2. The description of **critical-step detection** in DSS lacks clarity; how “critical timesteps” are selected or tuned is somewhat underexplained.
3. Most experiments emphasize efficiency, but the **effect on temporal consistency and perceptual coherence** (especially in long video sequences) is not sufficiently analyzed.
4. While claiming wide applicability, **the evaluation scope is limited** to a few DiT-based models; tests on other architectures (e.g., UNet-based diffusion models) would better demonstrate universality.
5. The method requires **careful parameter tuning** (e.g., cache ratio P, skip interval N) that may vary across conditions, yet the paper provides limited guidance on how to set these hyperparameters.

**Questions:**

1. Discussions and contributions vs. existing methods, such as ToCA and Duca are expected.
2. What are the failure cases or limitations of the proposed method?
3. Ablation studies on the temporal consistency and perceptual coherence.

---

### Note · Authors · 2025-11-12

I have read and agree with the venue's withdrawal policy on behalf of myself and my co-authors.